# Comparative Analysis of Dietary and Supplemental Intake of Calcium and Vitamin D among Canadian Older Adults with Heart Disease and/or Osteoporosis in 2004 and 2015

**DOI:** 10.3390/nu15245066

**Published:** 2023-12-11

**Authors:** Hassan Vatanparast, Ginny Lane, Naorin Islam, Rashmi Prakash Patil, Mojtaba Shafiee, Susan J. Whiting

**Affiliations:** 1College of Pharmacy and Nutrition, University of Saskatchewan, Saskatoon, SK S7N 4Z2, Canada; naorin.islam@usask.ca (N.I.); rashmi.patil@usask.ca (R.P.P.); mojtaba.shafiee@usask.ca (M.S.); susan.whiting@usask.ca (S.J.W.); 2School of Public Health, University of Saskatchewan, Saskatoon, SK S7N 4Z2, Canada; 3Margaret Ritchie School of Family and Consumer Sciences, University of Idaho, Moscow, ID 83843, USA; vlane@uidaho.edu

**Keywords:** calcium, vitamin D, dietary intake, supplement, heart disease, osteoporosis, older adults, Canadian population

## Abstract

Despite the role of calcium and vitamin D in osteoporosis and heart disease, little research has examined changes in the intake of calcium and vitamin D among individuals with these conditions over time. Using data from the 2004 and 2015 Canadian Community Health Surveys, we investigated changes in dietary and supplemental intake of calcium and vitamin D among Canadian older adults aged ≥ 50 years, both with and without heart disease and/or osteoporosis, between 2004 and 2015. Notable declines in dietary calcium intake occurred, particularly among non-supplement users. Surprisingly, individuals with osteoporosis and heart disease, who are at higher nutritional risk, were less likely to use calcium supplements in 2015 compared to 2004. Among calcium supplement users, those with osteoporosis or both conditions experienced significant reductions in their usual calcium intake in 2015, with an increased proportion failing to meet recommended intake levels. Conversely, vitamin D supplement users experienced a substantial rise in vitamin D intake in 2015. In 2015, only a small proportion of supplement users did not meet the recommended vitamin D intake levels. These findings underscore the importance of public health initiatives to facilitate safe increases in calcium and vitamin D intake for older adults, particularly those with heart disease and osteoporosis.

## 1. Introduction

According to the World Health Organization, chronic diseases are the leading cause of mortality globally, responsible for approximately 63% of all annual deaths [1]. These chronic diseases encompass non-communicable diseases (NCDs), long-term mental health disorders such as schizophrenia and major depression, and certain communicable diseases like HIV/AIDS. The escalating prevalence of chronic diseases poses a significant challenge to healthcare systems worldwide, both in terms of mortality rates and healthcare expenditures [2]. In Canada, chronic diseases have become a growing concern, accounting for approximately 88% of total deaths and constituting approximately one-third of direct healthcare expenses [3,4].

Among these chronic conditions, heart disease and osteoporosis are particularly notable. Heart disease, encompassing a range of cardiovascular conditions, is a leading cause of death globally [5]. It is characterized by conditions like coronary artery disease, heart failure, and arrhythmias, each contributing a substantial burden on healthcare systems [5]. Osteoporosis, on the other hand, is a bone disease that leads to increased fracture risk due to decreased bone mass and the deterioration of bone tissue [6]. It is especially prevalent in the aging population and contributes significantly to morbidity and healthcare costs [6].

The frequent co-occurrence of heart disease and osteoporosis in older adults, along with their shared underlying mechanisms such as inflammation and hormonal changes, highlights the importance of exploring their interrelations [7,8]. The substantial socioeconomic burden imposed by these chronic diseases underscores the complexities involved in managing them within the Canadian healthcare system [9,10].

Accumulating evidence underscores the significance of diet, a modifiable risk factor, in both the development and management of heart disease and osteoporosis [11,12]. In the context of bone health, extensive research has primarily focused on specific dietary components, notably calcium and vitamin D [13,14]. In November 2010, the Institute of Medicine (IOM), in collaboration with the American and Canadian governments, established the most recent dietary reference intakes (DRIs) for calcium and vitamin D. These recommendations serve as vital guidelines for maintaining bone health [15,16]. Our previous research, utilizing data from the Canadian Health Measures Survey (CHMS), has highlighted that approximately one-quarter of Canadians fail to meet the recommended dietary allowance (600 to 800 IU for those aged ≥ 1 year) for vitamin D. Notably, the use of supplements has contributed positively to achieving better plasma vitamin D levels [17]. Furthermore, studies have shown that, even with supplement usage, the average daily calcium intake among Canadian adults, particularly older adults, remains below the recommended levels essential for optimal bone health. For instance, women and men aged 25 years or older have reported mean daily calcium intakes of 1038 ± 614 mg and 904 ± 583 mg, respectively, which fall short of the recommended levels [18].

Emerging evidence indicates that micronutrients such as calcium and vitamin D are not only crucial for bone health but also play a role in reducing the risk of developing heart disease [19,20]. However, the literature offers varying reports concerning the relationships between dietary calcium/vitamin D intake, calcium/vitamin D supplementation, and the risk of heart disease. In a stratified analysis, Larsson and colleagues observed an inverse association between dietary calcium intake and the risk of stroke in populations with a low to moderate average calcium intake. Conversely, they noted a positive association in populations with a high calcium intake [21]. Furthermore, while certain studies have reported a connection between high intake of supplemental calcium and an increased risk of heart disease [22,23], others have not provided substantial support for this hypothesis [24,25].

To the best of our knowledge, there is limited research examining changes in the dietary and supplemental intake of calcium and vitamin D among individuals with heart disease and/or osteoporosis over time. Therefore, using data from the 2004 and 2015 Canadian Community Health Surveys, our primary objective in this study was to investigate alterations in the dietary and supplemental intake of calcium and vitamin D among older Canadian adults, both with and without heart disease and/or osteoporosis, between 2004 and 2015. Given the growing body of evidence linking calcium supplement usage to an increased risk of cardiovascular events, we hypothesize that the total calcium intake has declined among older adults with heart disease and/or osteoporosis between 2004 and 2015. Conversely, in light of the evidence supporting the cardioprotective effects of vitamin D, we anticipate that the total vitamin D intake has increased among older adults with heart disease and/or osteoporosis over the same period.

## 2. Materials and Methods

### 2.1. Study Population

This study utilizes data from the Canadian Community Health Survey (CCHS) Nutrition data for the years 2004 and 2015. The CCHS is a cross-sectional health survey conducted in Canada, designed to gather comprehensive health information from Canadian residents. Data on food consumption were collected using a 24 h dietary recall technique, with both 2004 and 2015 surveys capturing information on two separate days (day 1 and day 2). The 24 h dietary recall portion of the CCHS employs a sophisticated computer-assisted interviewing tool known as the Automated Multiple-Pass Method (AMPM). This method is designed to assist respondents in more accurately recalling and reporting the foods they consumed within the 24 h period preceding the interview, spanning from midnight to midnight [26,27]. For the 2015 dataset, a total of 20,487 individuals provided detailed information regarding their food consumption and socio-demographic variables on day 1, while 7623 individuals reported similar information on day 2. In the case of the 2004 dataset, the sample size for day 1 was 35,107 individuals, with a weighted frequency of 31,030,722, and for day 2, it was 10,786 [26,27]. As this study involved analyzing de-identified secondary data from the CCHS obtained via the Research Data Centre (RDC) of Statistics Canada, it was exempt from ethics approval per policies governing secondary data analysis.

### 2.2. Analytical Sample

This study represents a nationally representative sample of Canadian individuals aged 50 years and older. The data used in this analysis were drawn from a total of 8,887,312 (weighted frequency) and 12,399,600 (weighted frequency) individuals from the CCHS surveys conducted in 2004 and 2015, respectively. To ensure the robustness and comparability of the dataset, we applied specific exclusion criteria. Individuals under the age of 50, pregnant or lactating women, those with exceptionally high nutrient intakes, individuals who did not report any food intake during the 24 h recall, and those reporting a daily caloric intake outside the range of 200–8000 kcal were excluded. These exclusion criteria were consistently applied to both the 2004 and 2015 survey data.

### 2.3. Dietary and Supplemental Calcium and Vitamin D

The survey participants were queried about various aspects of their dietary and supplement intake, including the type of food consumed, the quantity of food, meal timing, location of consumption, and the use of dietary supplements. In the 2015 survey, information on supplement consumption, including frequency and quantity, was collected specifically for the preceding 24 h. In contrast, the 2004 survey collected supplement intake data differently. The participants were asked about their supplement usage over the 30 days leading up to the day of the interview. This approach allowed us to derive detailed information regarding supplement frequency, duration, and the daily amount consumed over this 30-day period.

### 2.4. Calcium and Vitamin D Requirements

The dietary reference intakes of calcium and vitamin D used in this study were defined by the IOM Food and Nutrition Board [16]. For individuals aged 50 and over, the estimated average requirements (EAR) and upper tolerable levels (UL) for calcium are as follows: male, 51–70 years: EAR 800 mg, UL 2000 mg; female, 51–70 years: EAR 1000 mg, UL 2000 mg; male, 71 years or older: EAR 1000 mg, UL 2000 mg; and female, 71 years or older: EAR 1000 mg, UL 2000 mg. For vitamin D, the EAR is 10 µg, and the UL is 100 µg for individuals aged 50 years and above.

### 2.5. Chronic Diseases

In both the CCHS surveys conducted in 2004 and 2015, the participants aged 19 years and older were inquired about their current status regarding specific chronic health conditions, all of which were diagnosed by healthcare professionals. These conditions included high blood pressure, diabetes, heart disease, cancer, and osteoporosis. Notably, the query about osteoporosis was directed exclusively to respondents aged 50 years and older.

### 2.6. Definitions and Covariates

The categorical variables used in this study include osteoporosis—self-reported (yes, no), heart disease—self-reported (yes, no), age (50–70 years, ≥71 years), sex (male, female), ethnicity (non-indigenous, indigenous), immigrant (yes, no), calcium supplement user (yes, no), vitamin D supplement user (yes, no), weight status (normal weight, overweight, obese), smoker (yes, no), education (<secondary graduate, secondary graduate, some post-secondary graduate, university graduate), region of residence (Atlantic, Quebec, Ontario, Prairies, British Columbia), and household income (decile 1–2, 3–4, 5–6, 7–8, 9–10) unless otherwise specified. We also calculated the percentages of individuals who met the recommended food guide serving per day of four food groups (vegetables and fruit, grain products, milk and alternatives, meat and alternatives) based on Canada’s 2007 Food Guide recommendations [28].

### 2.7. Statistical Analyses

All statistical analyses were completed using SAS, version 9.4. To produce population-level estimates, Statistics Canada used weighting and bootstrapping weights as per the recommendations. To calculate the usual calcium and vitamin D intakes and the percentages of the population below the EAR and above the UL from food only and from both food and dietary supplements for ages 50–70 years and >70 years, the National Cancer Institute (NCI) method was used in CCHS 2004 and 2015 with the following covariates:(i)The sequence of the 24 h recall, categorized as day 1 or day 2,(ii)The day of the week when the 24 h recall data were obtained and categorized as weekday or weekend,(iii)Energy consumption from food during the 24 h recall period.

For the NCI estimation, SAS macros developed by the National Cancer Institute (Usual Dietary Intakes: SAS Macros for the NCI Method, 2018) were used with a slight modification to account for bootstrapping weights. More details about the NCI method and the SAS macros can be found on the NCI website [29]. To compare the 2004 and 2015 results, the absence of overlapping 95% confidence intervals concept was applied [30]. Alpha was set at 0.05. The values are represented as percentages (SE) and means (SE) where required. The association between both chronic conditions (i.e., osteoporosis and heart disease) and sociodemographic variables was evaluated separately for each disease using multiple logistic regression models. Descriptive analyses were also carried out. Permission to access and conduct an analysis of the CCHS 2004 and 2015 data was obtained from the Saskatchewan Research Data Center Program of Statistics Canada.

## 3. Results

The results are presented in the following subsections, including descriptive data and the results from the statistical modelling.

### 3.1. Sociodemographic Characteristics

Table 1 presents the sociodemographic characteristics of the Canadians aged 50 and over who reported having osteoporosis or heart disease in 2004 and 2015. In 2004, the prevalence of osteoporosis was 11.3%, which decreased to 10.6% in 2015. Similarly, the prevalence of heart disease was 10.9% in 2004 and 9.3% in 2015. In 2015, a significant, 4.6% reduction in the proportion of female individuals with osteoporosis was reported compared to 2004. While there were no significant differences in vitamin D supplement use among individuals with osteoporosis between 2004 and 2015, there was a significant decrease in calcium supplement use in 2015. In 2004, 56.5% of individuals with osteoporosis used calcium supplements, compared to only 45.2% in 2015. The prevalence of osteoporosis significantly increased among smokers in 2015 compared to 2004, with rates of 12.5% in 2004 and 20.2% in 2015. Regarding education levels, the prevalence of osteoporosis significantly decreased among individuals with less than secondary education and university graduates. Conversely, it increased significantly among secondary graduates and post-secondary graduates in 2015 compared to 2004. A significant reduction was observed in the percentage of osteoporosis patients among those who met the recommended serving sizes for “grains” (5% decrease in 2015) and “meat and alternatives” (6% decrease in 2015) according to the Canada Food Guide 2007. Significant changes were also noted across income deciles. The lowest income group (decile 1–2) experienced a 27% increase in the prevalence of osteoporosis in 2015; the highest increase among all income categories. Additionally, the second-lowest income group (decile 3–4) saw an approximately 10% increase. However, the remaining middle- and high-income groups all experienced significant decreases in 2015.

A similar pattern was observed among heart disease patients and sociodemographic variables. In 2015, there was a significant decrease of approximately 8.5% in the percentage of heart disease patients taking calcium supplements compared to 2004. Regarding education levels, significant decreases were found in the proportion of heart disease patients among individuals with less than secondary education and university graduates, with reductions of 15.1% and 20.6%, respectively. In contrast, significant increases were observed among secondary graduates and some post-secondary graduates, with increases of 10.6% and 25.1%, respectively. A reduction in the percentage of heart disease patients was also noted among individuals who consumed the recommended servings of “vegetables and fruit” and “milk and alternatives” in 2015, with decreases of 7% and 5.3%, respectively. Similar to the pattern observed for osteoporosis, there were significant increases in the prevalence of heart disease among those in the lowest income groups, specifically decile 1–2 (25%) and 3–4 (16%). Conversely, the middle- and high-income groups experienced significant decreases in the proportion of heart disease patients, with reductions ranging from 8% to 18%.

### 3.2. Factors Associated with Osteoporosis and Heart Disease among Canadians Aged 50 and Older in 2004 and 2015

Table 2 presents the results of a multivariate logistic regression analysis examining the association between osteoporosis/heart disease and sociodemographic variables among the Canadian population aged 50 years and older. This includes individuals with and without these conditions, reflecting a comprehensive overview of the prevalence and associated factors across this demographic segment for the years 2004 and 2015. In 2004, individuals aged ≥71 years had 2.3 times higher odds of having osteoporosis, while females had 8.1 times higher odds compared to males. Additionally, individuals who consumed vitamin D supplements exhibited 1.9 times higher odds of having osteoporosis. Regarding income, those in the lowest income group (decile 1–2) were 2.9 times more likely to have osteoporosis than those in the highest income decile (9–10). Similarly, individuals in the middle-income groups (deciles 3–4, 5–6, and 7–8) were 1.7 times more likely to have osteoporosis. No significant effects were observed for ethnicity, immigration status, calcium supplement use, smoking status, region of residence, education, weight status, or food group consumption.

In 2015, the odds of having osteoporosis were 2.3 times higher among older adults (≥71 years) compared to adults aged 50–70 years, and they were 4.6 times higher among females compared to males. Individuals in the lowest income group (decile 1–2) and middle-income group (decile 5–6) were 2.4 times and 2.5 times more likely to have osteoporosis, respectively, compared to the highest income group (decile 9–10). No significant effects were observed for ethnicity, immigration status, vitamin D and calcium supplement use, smoking status, region of residence, education, weight status, or food group consumption.

In 2004, the individuals in the oldest age group (≥71 years) were 3.1 times more likely to report having heart disease compared to individuals aged 50–70 years, and males were 1.7 times more likely to have heart disease compared to females. Individuals in the middle-income deciles of 3–4 and 5–6 were 2.7 times and 2 times more likely to have heart disease compared to individuals in the highest income decile 9–10. The likelihood of having heart disease was not significantly affected by ethnicity, immigration status, calcium supplement use, vitamin D supplement use, smoking status, region of residence, education, weight status, or food group consumption in 2004.

In 2015, the odds of having heart disease were 3.4 times higher among individuals aged ≥ 71 years compared to the younger group, and they were 1.6 times higher among males compared to females. Residents of the Atlantic region were 1.6 times more likely to have heart disease compared to individuals who lived in other regions. Moreover, ethnicity, immigration status, vitamin D and calcium supplement use, smoking status, income, education, weight status, and food group consumption did not have any significant effect on heart disease reporting in 2015.

### 3.3. Dietary and Supplemental Intake of Calcium among Individuals with Heart Disease, Osteoporosis, or Both Conditions in 2004 and 2015, Comparing Supplement Users and Non-Users

Table 3 presents the mean usual calcium intake among Canadians who reported having heart disease, osteoporosis, or both conditions using the NCI method in both 2004 and 2015. The percentage of individuals reporting both chronic conditions was 1.8% in 2004 and 1.7% in 2015. Among the individuals who did not take calcium supplements, there was a significant decrease in usual calcium intake from food, with a reduction of approximately 78 mg for those with heart disease and a similar decrease for those without heart disease between 2004 and 2015. Similar trends were observed among non-supplement users with and without osteoporosis, where usual calcium intake from food significantly decreased by 68 to 75 mg in 2015 compared to 2004. Among the calcium supplement users, we observed significant decreases in the usual calcium intake among individuals with osteoporosis (302 mg) and those with both osteoporosis and heart disease (205 mg).

### 3.4. Prevalence of Calcium Inadequacy among Individuals with Heart Disease, Osteoporosis, or Both Conditions in 2004 and 2015, Comparing Supplement Users and Non-Users

Figure 1 illustrates the prevalence of calcium inadequacy among Canadians who reported having heart disease, osteoporosis, or both conditions in 2004 and 2015. Among the non-supplement users, the percentage of individuals who did not meet the estimated average requirement for calcium significantly increased by 5.6% for those with heart disease, 6.8% for those without heart disease, and 5.8% for those without osteoporosis in 2015 compared to 2004. For the supplement users, there was a noteworthy increase in the percentage of individuals who did not meet the estimated average requirement for calcium, with a 12% increase among those with osteoporosis and a 15% increase among those with both diseases between 2004 and 2015.

### 3.5. Dietary and Supplemental Intake of Vitamin D among Individuals with Heart Disease, Osteoporosis, or Both in 2004 and 2015, Comparing Supplement Users and Non-Users

Vitamin D intake from food remained relatively stable over the 11-year period (from 2004 to 2015) for all the groups, as shown in Table 4. However, across all categories of supplement users, there was a substantial increase in usual vitamin D intake, from both food and supplements, reaching approximately 39 µg in 2015, compared to the range of 16 to 19 µg in 2004.

### 3.6. Prevalence of Vitamin D Inadequacy among Individuals with Heart Disease, Osteoporosis, or Both in 2004 and 2015, Comparing Supplement Users and Non-Users

Figure 2 provides an overview of the prevalence of vitamin D inadequacy among supplement users across all the disease groups in 2004 and 2015. Notably, there was a significant decrease in the proportion of supplement users who did not meet the estimated average requirement for vitamin D, which dropped from 16–19% in 2004 to 6–7% in 2015 among all the disease groups.

## 4. Discussion

Between 2004 and 2015, there was a notable increase in osteoporosis and heart disease prevalence, particularly among the lowest income groups. Among those not taking calcium supplements, calcium intake from food decreased significantly in all the groups except for those with both diseases. Additionally, the proportion of non-supplement users not meeting their calcium requirements increased for those with and without heart disease and osteoporosis. Regarding calcium supplementation, both osteoporosis and heart disease patients were less likely to take calcium supplements in 2015 compared to 2004. Among the supplement users, those with osteoporosis or both diseases had lower calcium intake in 2015, and more individuals did not meet their calcium requirements. While vitamin D intake from food remained unchanged from 2004 to 2015 across all the groups, a significant portion of non-supplement users did not meet their vitamin D requirements. However, the supplement users had increased vitamin D intake in 2015, and fewer failed to meet the requirements during this period.

The observed decrease in dietary calcium consumption between 2004 and 2015 may suggest a decline in overall diet quality among adults aged 50 years and older or a deliberate avoidance of calcium-rich foods in response to confusing media reports about studies indicating cardiovascular health risks associated with high calcium intake, such as the study by Bolland et al. (2008) [31]. This aligns with our previous work, which indicated a decrease in the usual intake of calcium from food sources in calcium supplement nonusers across the general Canadian population aged ≥1 year during the same period [32]. Similarly, the findings regarding reduced use of calcium supplements by individuals with osteoporosis and heart disease, as well as the higher likelihood of supplement users with osteoporosis or both diseases not meeting the estimated average requirement for calcium in 2015 compared to 2004, may indicate that older adults with health conditions potentially benefiting from adequate calcium intake were dissuaded from consuming sufficient supplemental calcium to meet recommended intakes due to the influence of studies like Bolland et al. (2008) [31].

The Bolland et al. (2008) study was the first to report a potential negative effect of calcium supplementation on the risk of cardiovascular disease [31]. In this study, the participants had an average calcium intake of 1900 mg/day, with some individuals consuming amounts exceeding the upper limit, which raised concerns about the risk of hypercalcemia and vascular calcification of preexisting lesions. Excessive calcium intake above the upper limit can lead to hypercalcemia and vascular calcification, increasing the risk of cardiovascular issues [33]. Critics have pointed out several limitations in the Bolland et al. (2008) study [31], including that it was not primarily designed to investigate the relationship between calcium and cardiovascular disease, its small sample size, potential data misinterpretation, the use of a composite cardiovascular endpoint that obscured the actual impact of calcium supplementation on myocardial infarction, the failure to control for potential confounding factors, and limited generalizability due to the characteristics of the study sample [34]. Upon reanalysis of the Bolland et al. (2008) data using only cardiovascular event data from hospital records, no explicit adverse cardiovascular risk associated with calcium supplementation was found [34]. It has been estimated that a study would require 20,000 participants over 5 years to detect a 20% difference in cardiovascular risk related to calcium supplementation [35].

Following their 2008 publication, Bolland and colleagues conducted a meta-analysis that reaffirmed their initial findings [36]. However, it is important to note that this meta-analysis excluded trials of combined calcium and vitamin D supplementation, which significantly restricted the applicability of their conclusions. This limitation is significant, because the recommended approach to managing osteoporosis involves ensuring adequate intake of both calcium and vitamin D [37]. Subsequently, several other studies expressing concerns about the relationship between calcium intake and cardiovascular health have been published [22,38,39]. However, it is worth acknowledging that these studies faced criticisms, including issues related to selective data reporting and inconsistent interpretation, as highlighted in various letters to the editors.

Several recent meta-analyses have provided evidence suggesting that calcium intake within recommended levels does not pose an increased risk of cardiovascular events. One meta-analysis, which focused on prospective observational studies examining dietary calcium intake and cardiovascular disease risk, found that individuals with the highest dietary calcium intake had a pooled relative risk of 0.92 for any coronary artery disease (CAD) and 0.86 for any stroke when compared to those with the lowest intake [40]. Another meta-analysis, which included randomized controlled trials of calcium supplementation with and without vitamin D (including the Bolland et al. 2008 study), reported a pooled relative risk of 1.14 for cardiovascular events among individuals taking calcium supplements compared to placebos. However, the same meta-analysis found that those taking both calcium and vitamin D supplements had a pooled relative risk of 0.99 for cardiovascular events [41]. A meta-analysis conducted by Lewis et al. in 2015 also yielded similar results. They reported a pooled relative risk of 1.02 for coronary heart disease among individuals taking calcium supplementation with or without vitamin D when compared to those not taking calcium supplements. However, the same analysis found a pooled relative risk of 0.96 for all-cause mortality among individuals taking calcium supplementation with or without vitamin D compared to those not taking calcium supplements [42]. These findings, supported by other reviews, collectively suggest that maintaining dietary and supplemental calcium intake within recommended levels does not appear to increase cardiovascular risk and may not provide significant cardiovascular benefits for generally healthy adults [43,44].

A substantial body of literature has highlighted the beneficial effects of dietary and/or supplemental calcium on cardiovascular health, which includes improvements in cholesterol profiles [45], reductions in blood pressure [46], enhanced insulin sensitivity [47], and improved vasorelaxation [45]. Furthermore, recent evidence-based dietary reference intakes continue to recommend adequate calcium and vitamin D intake from food and, when necessary, supplements, particularly for older adults who may be at risk for osteoporosis and fractures [16]. For instance, a study conducted in postmenopausal women with osteoporosis in France revealed that the average daily dietary vitamin D intake was only 144.8 IU, with 30% of the participants taking a vitamin D supplement. The average daily dietary calcium intake was 966.4 mg, with 38% of the participants using a calcium supplement [48]. In another study among Spanish females aged 50 years and older with osteoporosis, the mean dietary calcium intake was 1239 mg/day, whereas those aged 75 years and older had a significantly lower mean dietary calcium intake of 988 mg/day. Dietary vitamin D intake was reported at 167 IU/day, with even lower intake among participants aged 75 years and older (120 IU/day) [49].

The decline in calcium consumption among older Canadians during a period when the media was highlighting potential cardiovascular risks linked to calcium intake, as indicated in various scientific publications, underscores the importance of responsible research dissemination. Drawing broad conclusions from studies can create unwarranted alarm and may lead the public to discontinue healthy dietary habits, potentially putting them at greater nutritional risk. We have a moral obligation not only to conduct ethical research but also to ethically communicate our findings, with the ultimate goal of safeguarding public health.

Our findings indicate a significant increase in both the prevalence of vitamin D supplement use and the doses of supplemental vitamin D among Canadian adults aged 50 years and older from 2004 to 2015. This aligns with our previous work on vitamin D intake in the general Canadian population aged ≥1 year, which showed that, while vitamin D intake from food remained largely unchanged during this period, the prevalence of vitamin D supplement use and the percentage contribution of vitamin D from supplemental sources increased significantly [50]. This upward trend in vitamin D supplement usage and doses could be attributed to the growing body of literature suggesting that vitamin D supplementation may have a protective effect against cardiovascular disease (CVD) [51,52,53]. For instance, a meta-analysis of eight prospective cohort studies from the United States and Europe by Schöttker et al. found that individuals in the lowest quintile of serum 25(OH)D concentration had an increased risk of cardiovascular and all-cause mortality [54]. However, it is worth noting that while epidemiological evidence [55] and observational studies [56,57,58,59] have shown an association between vitamin D deficiency and CVD risk, this has not been confirmed by randomized controlled trials [60]. Another meta-analysis of 21 randomized controlled trials found that vitamin D supplementation did not reduce major adverse cardiovascular events, specific CVD endpoints, or all-cause mortality [61]. Furthermore, concerning bone health and osteoporosis, a recent randomized controlled trial led by Michos et al. discovered that achieving a 25(OH)D concentration of ≥40 ng/mL after vitamin D supplementation in adults aged ≥70 was not associated with a reduction in falls; in fact, it was associated with an increased risk of consequential falls [62]. It is important to recognize that low dietary vitamin D does not invariably signal a deficiency due to the body’s capacity to synthesize it through skin exposure to ultraviolet B (UVB) rays, a process that is subject to seasonal variations in regions like Canada. This consideration is essential when interpreting the non-linear relationship between serum 25(OH)D concentration and health outcomes, which can be influenced by various sources, including diet, UVB exposure (particularly during spring and summer), and supplementation. This multifaceted interplay may explain why randomized controlled trials often fail to confirm findings from observational studies [63]. Additionally, it is worth mentioning that the upper limit recommended by the National Institutes of Health (NIH) and Institute of Medicine (IOM) is 4000 IU daily for individuals aged 9 years and older, underscoring the safe threshold for supplementation [16,64].

Strengths and limitations: In terms of strengths, this study draws on data from two extensive, nationally representative surveys, encompassing both dietary and supplemental intake information. This comprehensive approach enhances the study’s credibility and generalizability to the Canadian population. However, there are notable limitations to consider. Firstly, the study primarily examines changes in calcium and vitamin D consumption among older age groups, and it would be valuable to explore sex-disaggregated data to unveil potential gender-specific trends. Additionally, variations in survey execution between survey cycles, including adjustments to the food booklet used for estimating portion sizes and changes in nutrient databases, may impact the comparability of dietary intake data. Specifically, the shift from line drawings to actual-size photographs in 2015 could introduce a downward bias, particularly affecting estimations of energy, tea, and coffee consumption [65]. Furthermore, the information regarding calcium and vitamin D supplement use was collected differently in 2015, covering the past 24 h, compared to 2004, which included data from the previous 30 days. These reporting period differences should be considered when interpreting the findings. Lastly, there might be some overlap between vitamin D and calcium supplement users, since many supplements contain both nutrients, potentially influencing the study’s results. These limitations should be taken into account when interpreting the study’s outcomes and planning future research in this area.

## 5. Conclusions

In conclusion, this study reveals a concerning decline in the usual calcium intake from food among older Canadians between 2004 and 2015. Moreover, a substantial proportion of non-supplement users did not meet the estimated average requirement for calcium, highlighting a nutritional gap. Additionally, among the calcium supplement users, those with osteoporosis or both osteoporosis and heart disease were more likely to fall below the estimated average requirement for calcium in 2015 compared to 2004, indicating potential risk reduction behaviors. Vitamin D inadequacy was also prevalent, with a majority of non-supplement users failing to meet the estimated average requirement in both 2004 and 2015. However, the supplement users demonstrated improved vitamin D intake. The implications of these findings are particularly relevant in light of the recent changes to the Canadian food guidelines, which no longer emphasize dairy as a distinct food group for mandatory consumption. This change may impact calcium intake among those who may reduce or eliminate dairy consumption without adequately replacing it with other calcium-rich foods or supplements. Therefore, it is imperative for public health initiatives to focus on older adults, promoting a balanced intake through both diet and supplementation where needed to bolster heart and bone health. Further research with a longitudinal design is needed to corroborate these results and strengthen this study’s conclusions.

## Figures and Tables

**Figure 1 nutrients-15-05066-f001:**
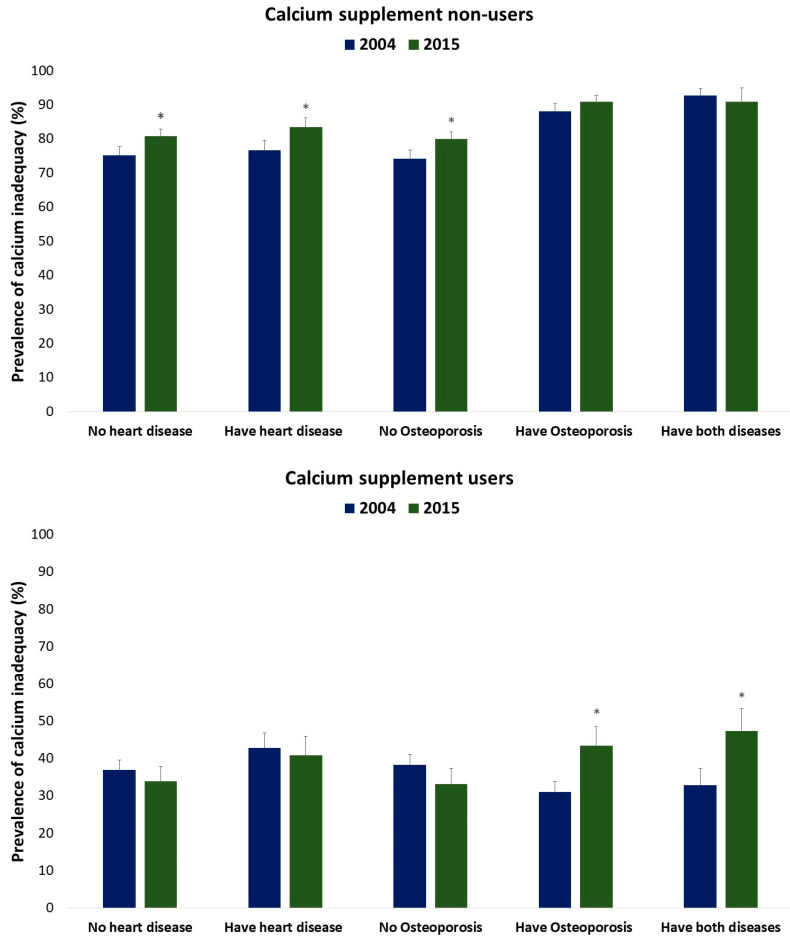
Prevalence of calcium inadequacy among individuals who had heart disease, osteoporosis, or both in 2004 and 2015, comparing supplement users and non-users. Calcium inadequacy was defined as an intake level below the EAR of 800 mg for males aged 51–70 years, 1000 mg for females aged 51–70 years, and 1000 mg for all individuals over 70 years, as specified by the IOM. The analysis was conducted separately for the 2004 and 2015 data sets. * Significant difference (*p* < 0.05) in supplement users between 2004 and 2015 based on age/sex groups using the confidence interval overlap technique.

**Figure 2 nutrients-15-05066-f002:**
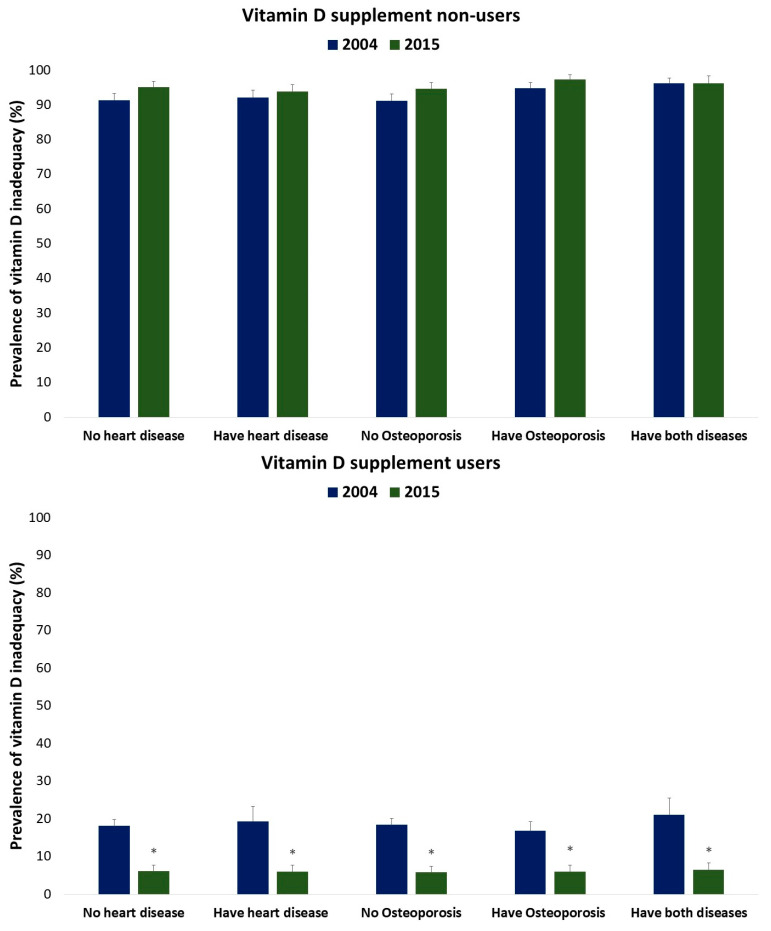
Prevalence of vitamin D inadequacy among individuals who had heart disease, osteoporosis, or both in 2004 and 2015, comparing supplement users and non-users. Vitamin D inadequacy was identified when intake levels were below the EAR of 10 µg (400 IU) for individuals aged 50 years and above, following the IOM recommendations. The analysis was conducted separately for the 2004 and 2015 data sets. * Significant difference (*p* < 0.05) in supplement users between 2004 and 2015 based on age/sex groups using the confidence interval overlap technique.

**Table 1 nutrients-15-05066-t001:** Sociodemographic descriptions of individuals with osteoporosis and heart disease in 2004 and 2015 among Canadians 50 years and older ^1^.

Sociodemographic Variables	Osteoporosis				Heart Disease			
	2004 (n = 1,061,034) (11.3%)		2015 (n = 1,375,730) (10.6%)		2004 (n = 1,028,909) (10.9%)		2015 (n = 1,211,896) (9.3%)	
	% ± SE	95% CI	% ± SE	95% CI	% ± SE	95% CI	% ± SE	95% CI
Age								
50–70 years	54.1 ± 2.0	(50.2, 58.1)	55.4 ± 2.4	(50.6, 60.1)	48.0 ± 2.2	(43.4, 52.6)	48.6 ± 2.9	(42.9, 54.4)
≥71 years	45.9 ± 2.0	(42.0, 49.8)	44.6 ± 2.4	(39.9, 49.4)	52.0 ± 2.2	(46.8, 56.0)	51.4 ± 2.9	(45.6, 57.1)
Sex								
Male	11.0 ± 1.4	(8.2, 13.8)	15.6 ± 1.8 ^¥^	(12.1, 19.2)	57.4 ± 2.1	(53.4, 61.5)	55.6 ± 2.9	(49.9, 61.3)
Female	89.0 ± 1.4	(86.2, 91.8)	84.4 ± 1.8 ^¥^	(80.8, 88.0)	42.6 ± 2.1	(38.5, 46.6)	44.4 ± 2.9	(38.7, 50.1)
Ethnicity								
Non-indigenous	98.6 ± 0.4	(97.7, 99.4)	97.3 ± 0.8	(95.7, 99.0)	97.9 ± 0.5	(97.0, 98.9)	97.7 ± 0.8	(96.2, 99.3)
Indigenous	1.5 ± 0.4	(0.6, 2.3)	2.7 ± 0.8	(1.0, 4.3)	2.1 ± 0.5	(1.1, 3.0)	2.3 ± 0.8	(0.7, 3.8)
Immigrant								
Yes	27.3 ± 2.3	(22.5, 31.5)	24.0 ± 4.4	(18.8, 29.3)	20.6 ± 1.9	(16.9, 24.3)	24.0 ± 2.8	(18.5, 29.4)
No	73.3 ± 2.3	(68.5, 77.5)	76.0 ± 3.3	(70.8, 81.2)	79.5 ± 1.9	(75.8, 83.2)	76.0 ± 2.8	(70.6, 81.5)
Calcium Supplement user								
Yes	56.5 ± 1.2	(52.1, 60.8)	45.2 ± 4.4 ^¥^	(39.2, 51.2)	33.9 ± 1.9	(29.5, 38.2)	24.5 ± 2.6 ^¥^	(19.5, 29.6)
No	43.5 ± 0.2	(39.2, 47.9)	54.8 ± 3.3 ^¥^	(48.8, 60.8)	66.2 ± 0.2	(61.8, 70.5)	75.5 ± 2.6 ^¥^	(70.4, 80.5)
Vitamin D Supplement user								
Yes	55.1 ± 1.3	(50.7, 59.5)	56.5 ± 4.4	(50.5, 62.5)	32.9 ± 1.2	(28.7, 37.2)	38.0 ± 2.9	(32.2, 43.7)
No	44.9 ± 0.3	(40.5, 49.4)	43.5 ± 3.3	(37.5, 49.5)	67.1 ± 0.2	(62.8, 71.3)	62.0 ± 2.9	(56.3, 67.8)
Weight Status								
Normal weight	39.7 ± 4.3	(34.4, 45.0)	37.5 ± 4.7	(30.3, 44.8)	26.7 ± 4.2	(21.3, 32.1)	28.0 ± 3.5	(21.1, 34.8)
Overweight	38.5 ± 5.3	(33.3, 43.7)	38.9 ± 5.1	(31.0, 46.9)	43.3 ± 5.4	(37.2, 49.5)	42.7 ± 4.3	(34.3, 51.0)
Obese	21.8 ± 6.2	(16.8, 26.8)	23.5 ± 6.8	(18.1, 29.0)	30.0 ± 6.3	(24.1, 35.9)	29.4 ± 3.9	(21.8, 37.0)
Smoking status								
Yes	12.5 ± 1.4	(9.9, 15.2)	20.2 ± 3.0 ^¥^	(14.3, 26.1)	18.6 ± 2.0	(14.6, 22.6)	15.6 ± 2.4	(10.9, 20.3)
No	87.5 ± 1.4	(84.8, 90.1)	79.8 ± 3.0 ^¥^	(73.9, 85.8)	81.4 ± 2.0	(77.5, 85.4)	84.4 ± 2.4	(79.8, 89.1)
Education								
<secondary graduate	41.7 ± 2.4	(37.1, 46.4)	26.3 ± 2.5 ^¥^	(21.5, 31.2)	44.7 ± 2.3	(40.2, 49.1)	29.6 ± 2.7 ^¥^	(24.4, 34.8)
secondary graduate	16.1 ± 1.8	(12.6, 19.5)	32.6 ± 3.0 ^¥^	(26.8, 38.4)	13.4 ± 1.4	(10.7, 16.2)	24.0 ± 2.5 ^¥^	(19.1, 28.9)
some post-secondary graduate	5.1 ± 0.9	(3.4, 6.8)	29.1 ± 2.7 ^¥^	(23.7, 34.5)	5.4 ± 0.9	(3.6, 7.2)	30.5 ± 3.1 ^¥^	(24.4, 36.6)
university graduate	37.1 ± 2.2	(32.8, 41.4)	12.0 ± 1.6 ^¥^	(8.8, 15.2)	36.5 ± 2.1	(32.3, 40.7)	15.9 ± 2.3 ^¥^	(11.3, 20.5)
Region of residence								
Atlantic	7.5 ± 0.8	(5.9, 7.7)	8.2 ± 0.8	(6.6, 8.8)	10.0 ± 0.9	(8.2, 10.9)	10.9 ± 1.0	(9.0, 12.9)
Quebec	24.3 ± 2.2	(20.0, 24.2)	24.7 ± 2.6	(19.6, 24.7)	24.4 ± 2.2	(20.1, 24.6)	29.7 ± 3.0	(23.7, 35.7)
Ontario	38.6 ± 2.0	(34.7, 38.4)	41.1 ± 2.8	(35.5, 41.6)	36.6 ± 1.9	(33.0, 36.3)	35.5 ± 3.0	(29.5, 41.4)
Prairies	16.6 ± 1.3	(14.1, 16.1)	15.1 ± 1.6	(12.0, 15.3)	13.6 ± 1.2	(11.3, 13.0)	14.3 ± 1.6	(11.1, 17.4)
British Columbia	13.1 ± 1.4	(10.3, 13.1)	10.9 ± 1.5	(8.0, 10.9)	15.3 ± 1.7	(12.0, 15.6)	9.7 ± 1.5 ^¥^	(6.7, 12.6)
Met daily grain product requirement								
Yes	23.2 ± 2.2	(18.2, 27.5)	18.0 ± 2.4 ^¥^	(13.4, 22.6)	23.6 ± 2.2	(19.2, 28.0)	19.5 ± 2.6	(14.4, 24.6)
No	76.8 ± 2.2	(72.8, 81.1)	82.0 ± 2.4 ^¥^	(77.4, 86.6)	76.4 ± 2.2	(72.0, 80.8)	80.5 ± 2.6	(75.5, 85.6)
Met daily meat and alternatives requirement								
Yes	30.7 ± 2.2	(26.4, 35.0)	24.5 ± 4.6 ^¥^	(19.4, 29.6)	23.1 ± 1.9	(19.4, 26.8)	20.6 ± 2.7	(15.2, 25.9)
No	69.3 ± 2.2	(65.0, 73.6)	75.5 ± 3.6 ^¥^	(70.4, 80.6)	76.9 ± 1.9	(73.2, 80.6)	79.4 ± 2.7	(74.1, 84.8)
Met daily milk and alternatives requirement								
Yes	9.3 ± 1.0	(7.3, 11.3)	11.0 ± 1.7	(7.7, 14.4)	14.9 ± 1.7	(11.5, 18.3)	9.6 ± 1.7 ^¥^	(6.3, 13.0)
No	90.7 ± 1.0	(88.7, 92.7)	89.0 ± 1.7	(85.6, 92.3)	85.1 ± 1.7	(81.7, 88.5)	90.4 ± 1.7 ^¥^	(87.1, 93.7)
Met daily vegetable and fruit requirement								
Yes	20.9 ± 1.8	(17.4, 24.3)	16.9 ± 2.6	(11.9, 22.0)	21.8 ± 1.9	(18.1, 25.4)	14.8 ± 2.2 ^¥^	(10.4, 19.1)
No	79.1 ± 1.8	(75.7, 82.6)	83.1 ± 2.6	(78.0, 88.1)	78.2 ± 1.9	(74.6, 81.9)	85.2 ± 2.2 ^¥^	(80.9, 89.6)
Income decile								
Decile 1–2	6.2 ± 1.8	(2.7, 6.7)	33.3 ± 3.1 ^¥^	(27.2, 39.5)	3.3 ± 0.7	(1.8, 4.7)	28.0 ± 2.5 ^¥^	(23.1, 32.9)
Decile 3–4	11.4 ± 1.4	(8.6, 11.3)	21.1 ± 2.0 ^¥^	(17.1, 25.0)	13.5 ± 1.6	(10.4, 16.6)	29.3 ± 3.0 ^¥^	(23.5, 35.1)
Decile 5–6	30.5 ± 2.2	(26.2, 30.7)	23.2 ± 2.5 ^¥^	(18.2, 28.2)	36.3 ± 2.3	(31.7, 40.8)	22.2 ± 2.7 ^¥^	(16.9, 27.4)
Decile 7–8	34.7 ± 2.5	(29.9, 34.6)	14.0 ± 2.4 ^¥^	(9.4, 18.7)	28.8 ± 2.2	(24.6, 33.1)	10.8 ± 1.7 ^¥^	(7.5, 14.1)
Decile 9–10	17.2 ± 2.1	(13.1, 17.2)	8.4 ± 1.6 ^¥^	(5.3, 11.5)	18.2 ± 2.0	(14.4, 22.0)	9.7 ± 2.3 ^¥^	(5.1, 14.3)

^1^ Values are mean ± SE, and the analyses were conducted separately for the 2004 and 2015 data sets. ^¥^ Significant differences (*p*-value < 0.05) between CCHS 2004 and 2015 using the confidence interval overlapping technique.

**Table 2 nutrients-15-05066-t002:** Factors associated with osteoporosis and heart disease among Canadians (≥50 years) in 2004 and 2015 ^1^.

	Osteoporosis		Heart Disease	
	2004 (n = 1,061,034) (11.3%)	2015 (n = 1,375,730) (10.6%)	2004 (n = 1,028,909) (10.9%)	2015 (n = 1,211,896) (9.3%)
	OR (95% CI)	OR (95% CI)	OR (95% CI)	OR (95% CI)
Age				
50–70 years	1	1	1	1
≥71 years	2.3 * (1.8, 2.9)	2.3 * (1.7, 3.1)	3.1 * (2.4, 4)	3.4 * (2.4, 4.8)
Sex				
Male ^2^	1	1	1.7 * (1.3, 2.2)	1.6 * (1.2, 2.2)
Female ^3^	8.1 * (5.5, 12)	4.6 * (3.1, 6.6)	1	1
Ethnicity				
Indigenous ^4^	1	1	1	1
Non-indigenous	0.6 (0.2, 1.6)	0.6 (0.3, 1.3)	0.8 (0.4, 1.8)	0.995 (0.4, 2.6)
Immigrant				
No ^4^	1	1	1	1
Yes	1.1 (0.8, 1.5)	0.8 (0.5, 1.3)	0.8 (0.6, 1)	0.9 (0.6, 1.3)
Calcium supplement User				
No ^4^	1	1	1	1
Yes	1.1 (0.8, 1.6)	1.2 (0.8, 1.7)	1 (0.6, 1.6)	0.8 (0.5, 1.3)
Vitamin D supplement User				
No ^4^	1	1	1	1
Yes	1.9 * (1.3, 2.8)	1.5 (1.0, 2.1)	0.9 (0.6, 1.5)	1.1 (0.7, 1.8)
Weight Status				
Normal weight ^4^	1	1	1	1
Overweight	0.9 (0.7, 1.2)	1.1 (0.8, 1.6)	1.2 (0.9, 1.6)	1.1 (0.7, 1.6)
Obese	0.7 (0.5, 0.9)	0.9 (0.6, 1.3)	1.4 (1, 1.9)	1.5 (1, 2.2)
Smoking status				
No ^4^	1	1	1	1
Yes	0.8 (0.6, 1.1)	1.6 (1, 2.5)	1.04 (0.7, 1.5)	1.1 (0.7, 1.8)
Education				
University graduate ^4^	1	1	1	1
<Secondary graduate	1.3 (1, 1.8)	1.6 (1, 2.6)	1.2 (0.9, 1.7)	1.1 (0.7, 1.7)
Secondary graduate	0.9 (0.6, 1.3)	1.2 (0.8, 1.8)	1 (0.7, 1.4)	0.8 (0.5, 1.1)
Some post-secondary graduate	1.1 (0.7, 1.8)	1.7 (1, 2.9)	0.9 (0.6, 1.5)	0.9 (0.6, 1.5)
Region of residence				
Ontario ^4^	1	1	1	1
Atlantic	1 (0.7, 1.4)	1 (0.7, 1.5)	0.9 (0.7, 1.3)	1.6 * (1.1, 2.5)
Quebec	1 (0.7, 1.5)	0.9 (0.6, 1.4)	0.8 (0.6, 1)	1.4 (0.9, 2.3)
Prairies	1.2 (0.9, 1.6)	0.9 (0.6, 1.3)	0.8 (0.6, 1)	1.2 (0.8, 1.8)
British Columbia	0.7 (0.5, 1.1)	0.8 (0.5, 1.2)	1.2 (0.8, 1.8)	0.9 (0.6, 1.5)
Income				
Decile 9–10 ^4^	1	1	1	1
Decile 1–2	2.9 * (1.2, 6.9)	2.4 * (1.3, 4.5)	1.7 (0.9, 3.4)	1.8 (0.9, 3.6)
Decile 3–4	1.7 * (1, 2.7)	1.6 (0.9, 2.8)	2.7 * (1.7, 4.2)	1.6 (0.8, 3.2)
Decile 5–6	1.7 * (1.2, 2.5)	2.5 * (1.3, 4.5)	2 * (1.4, 2.8)	1.5 (0.7, 3)
Decile 7–8	1.7 * (1.2, 2.4)	1.9 (1, 3.5)	1.4 (1, 1.9)	0.9 (0.5, 1.9)
Met daily grain product requirement				
No ^4^	1	1	1	1
Yes	1.1 (0.8, 1.6)	1.1 (0.7, 1.7)	1.3 (0.9, 1.8)	0.9 (0.6, 1.4)
Met daily meat and alternatives requirement				
No ^4^	1	1	1	1
Yes	1 (0.7, 1.3)	0.8 (0.6, 1.2)	0.8 (0.6, 1.1)	0.8 (0.5, 1.1)
Meet daily milk and alternatives requirement				
No ^4^	1	1	1	1
Yes	0.7 (0.5, 1.1)	1.2 (0.7, 2.1)	1.3 (0.9, 2)	1 (0.6, 1.7)
Meet daily vegetable and fruit requirement				
No ^4^	1	1	1	1
Yes	0.8 (0.6, 1.1)	1.2 (0.8, 1.9)	1 (0.7, 1.3)	1 (0.7, 1.6)

^1^ Values are OR (95% CI), and the analyses were conducted separately for the 2004 and 2015 data sets. ^2^ Reference group for osteoporosis. ^3^ Reference group for heart disease. ^4^ Reference category of the odds ratio. * Significant differences (*p*-value < 0.05) within categories.

**Table 3 nutrients-15-05066-t003:** Dietary and supplemental intake of calcium among Canadians (≥50 years) who had heart disease, osteoporosis, and both diseases combined among supplement users and non-users in 2004 and 2015 ^1^.

	Calcium Supplement Non-Users		Calcium Supplement Users			
	2004 (61.5%)	2015 (67.9%)	2004 (38.5%)		2015 (32.1%)	
Chronic Diseases	Calcium (mg) from Food	Calcium (mg) from Food	Calcium (mg) from Food	Calcium (mg) from Food and Supplements	Calcium (mg) from Food	Calcium (mg) from Food and Supplements
No heart disease	732.9 ± 30.0	654.8 ± 26.0 ^¥^	733.5 ± 33.5	1244.7 ± 37.6	738.4 ± 38.8	1193.9 ± 57.0
Have heart disease	720.4 ± 32.7	641.7 ± 34.6 ^¥^	657.8 ± 34.3	1169.4 ± 56.6	671.0 ± 41.5	1126.3 ± 57.7
No Osteoporosis	737.9 ± 30.1	662.7 ± 25.8 ^¥^	736.8 ± 34.0	1204.5 ± 38.1	744.1 ± 39.4	1203.3 ± 57.2
Have Osteoporosis	648.3 ± 31.9	580.6 ± 27.8 ^¥^	668.1 ± 34.0	1419.4 ± 46.2	649.6 ± 36.4	1117.6 ± 56.1 ^¥^
Have both diseases	594.8 ± 32.0	592.9 ± 47.8	637.9 ± 35.0	1337.9 ± 67.9	602.8 ± 47.0	1082.9 ± 61.1 ^¥^

^1^ Values are mean ± SE, and the analyses were conducted separately for the 2004 and 2015 data sets. ^¥^ Significant differences (*p*-value < 0.05) between CCHS 2004 and 2015 using the confidence interval overlapping technique.

**Table 4 nutrients-15-05066-t004:** Dietary and supplemental intake of vitamin D among Canadians (≥50 years) who had heart disease, osteoporosis, or both diseases combined among supplement users and non-users in 2004 and 2015 ^1^.

	Vitamin D Supplement Non-Users		Vitamin D Supplement Users			
	2004 (63.7%)	2015 (58.0%)	2004 (36.3%)		2015 (42.0%)	
Chronic Diseases	Vitamin D (µg) from Food	Vitamin D (µg) from Food	Vitamin D (µg) from Food	Vitamin D (µg) from Food and Supplements	Vitamin D (µg) from Food	Vitamin D (µg) from Food and Supplements
No heart disease	5.4 ± 0.3	4.7 ± 0.4	5.3 ± 0.5	16.6 ± 0.6	4.8 ± 0.5	39.6 ± 3.3 ^¥^
Heart Disease	5.3 ± 0.4	4.8 ± 0.4	4.8 ± 0.4	16.5 ± 1.2	4.3 ± 0.5	38.6 ± 3.3 ^¥^
No Osteoporosis	5.4 ± 0.3	4.8 ± 0.4	5.4 ± 0.5	16.1 ± 0.5	4.8 ± 0.5	39.9 ± 3.3 ^¥^
Have Osteoporosis	4.8 ± 0.3	4.2 ± 0.4	4.8 ± 0.4	19.3 ± 1.1	4.4 ± 0.4	38.9 ± 3.3 ^¥^
Have both diseases	4.6 ± 0.3	4.5 ± 0.5	4.4 ± 0.4	18.6 ± 3.3	4.0 ± 0.4	39.0 ± 3.2 ^¥^

^1^ Values are mean ± SE, and the analyses were conducted separately for the 2004 and 2015 data sets. ^¥^ Significant differences (*p*-value < 0.05) between CCHS 2004 and 2015 using the confidence interval overlapping technique.

## Data Availability

The data are available at the Research Data Centre (RDC) of Statistics Canada.

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
