# Peer review of "Comparative Analysis of Dietary and Supplemental Intake of Calcium and Vitamin D among Canadian Older Adults with Heart Disease and/or Osteoporosis in 2004 and 2015"

_nutrients, 2023, doi:10.3390/nu15245066_

Round 1

Reviewer 1 Report

Comments and Suggestions for Authors

Dear  Authors.

Please read the article carefully and make the necessary corrections and additions.

1.      The title suggests that the research was conducted continuously from 2004 to 2015, while the manuscript shows that the research concerned only 2004 and 2015. Please explain this.

2.      The introduction contains too little information about the diseases discussed, please elaborate.

3.      The first paragraph of the discussion is the results, please move this to the appropriate section.

Author Response

Reviewer 1:

Dear Authors.

Please read the article carefully and make the necessary corrections and additions.

  1. The title suggests that the research was conducted continuously from 2004 to 2015, while the manuscript shows that the research concerned only 2004 and 2015. Please explain this.

Response: Thank you for your thoughtful feedback on our manuscript. The title is intended to encapsulate the scope of our research, which investigates changes in intake patterns over the specified period. However, you are correct in noting that our analysis is not based on continuous data collection throughout these years. Instead, our research utilizes data points specifically from the years 2004 and 2015.

We acknowledge that the title could potentially lead to misinterpretation regarding the continuity of data collection. To more accurately reflect our methodology and address your concern, we propose the following revised title: "Comparative Analysis of Dietary and Supplemental Intake of Calcium and Vitamin D Among Canadian Older Adults with Heart Disease and/or Osteoporosis in 2004 and 2015"

This revised title explicitly states that the study is a comparative analysis between two specific years, thereby eliminating any potential confusion about the continuity of data collection over the entire period from 2004 to 2015.

  1. The introduction contains too little information about the diseases discussed, please elaborate.

Response: Thanks, we added some more information about heart disease and osteoporosis to the introduction. Page 1, lines 41-51.

  1. The first paragraph of the discussion is the results, please move this to the appropriate section.

Response: Thank you for your feedback regarding the structure of the manuscript, particularly the first paragraph of the discussion section. We understand your concern about the content appearing more like a summary of results than a discussion. In our approach, we typically begin the discussion section with a brief summary of the key findings as a way to set the stage for a more detailed discussion. This approach is intended to provide a quick recap of the results for readers before delving into the deeper analysis and implications of these findings. We believe that this helps in creating a smooth transition from the results section to the discussion, aiding readers in following the narrative flow of the paper. This approach aligns with recommended practices in scientific writing, as outlined in resources like the guidelines provided by the PLOS on structuring the discussion section of a paper (https://plos.org/resource/how-to-write-conclusions/).

However, we appreciate your perspective and are open to adjusting the structure to align more closely with the conventional format. If it is more in line with the journal's style and preferences, we can move this summary to the end of the results section or reframe it within the discussion to more explicitly connect the results to the broader context and implications.

Reviewer 2 Report

Comments and Suggestions for Authors

 The manuscript “Changes in Dietary and Supplemental Intake of Calcium and  Vitamin D Among Canadian Older Adults with Heart Disease  and/or Osteoporosis from 2004 to 2015”by Hassan Vatanparast et al  aimed to evaluate the intake of calcium and vitamind D in Canadian Older Adults.

 COMMENTS

1). The Authors must explain how they managed to evaluate the intake of vitamin D and calcium from foods. Further information on the proposed questionnaire is essential.

2). The fact that the assessment of calcium and vitamin D intake was assessed with very
different methods in 2004 and 2015 represents an important limitation of the study.
3). Table 1: The mean values (Mean +/- SD) of calcium and vitamin D intake with food

and supplementation in 2004 and 2015 must be reported in this table.
4). The Authors must explain whether the diagnosis of "Osteoporosis" and "Heart Disease" is based only on the participants' declarations or confirmed by the analysis of health documentation. 5). Figures 1 and 2: In the legends to Figures 1 and 2, the threshold values of calcium intake and vitamin intake that identify a condition of "inadequacy" must be reported.

6). Approval of the epidemiological study by the Ethics Committee should be reported.

Author Response

Reviewer 2:

The manuscript “Changes in Dietary and Supplemental Intake of Calcium and  Vitamin D Among Canadian Older Adults with Heart Disease  and/or Osteoporosis from 2004 to 2015”by Hassan Vatanparast et al  aimed to evaluate the intake of calcium and vitamind D in Canadian Older Adults.

COMMENTS

1) The Authors must explain how they managed to evaluate the intake of vitamin D and calcium from foods. Further information on the proposed questionnaire is essential.

Response: Thank you for your thoughtful feedback on our manuscript. We added more information on the structure of the 24-hour dietary recall questionnaire, which was employed to gather data on food consumption. Page 3, lines 100-103.

2) The fact that the assessment of calcium and vitamin D intake was assessed with very
different methods in 2004 and 2015 represents an important limitation of the study. 

Response: We appreciate your attention to the methodologies employed in our study for assessing calcium and vitamin D intake across the 2004 and 2015 surveys. To clarify, both surveys utilized a consistent method for collecting dietary data – the 24-hour dietary recall technique known as the Automated Multiple-Pass Method (AMPM). The AMPM is a standardized and validated method for dietary assessment, and its use across both survey periods ensures a consistent approach in data collection. This method involves a detailed and structured interview process that prompts respondents to recall and report all foods and beverages consumed in the 24-hour period preceding the interview. The use of AMPM in both 2004 and 2015 allowed for a reliable and comparable assessment of dietary intake, including nutrients like calcium and vitamin D. Page 3, lines 98-108.

3) Table 1: The mean values (Mean +/- SD) of calcium and vitamin D intake with food and supplementation in 2004 and 2015 must be reported in this table.

Response: Thank you for your comment. The prevalence of calcium and vitamin D supplement use is presented in Table 1. Furthermore, the mean dietary intake data of calcium and vitamin D, including both food and supplementation for 2004 and 2015, are detailed in Table 3.

4) The Authors must explain whether the diagnosis of "Osteoporosis" and "Heart Disease" is based only on the participants' declarations or confirmed by the analysis of health documentation.

Response: Thank you for your inquiry regarding the basis of the diagnoses of "Osteoporosis" and "Heart Disease" in our study participants. As stated in our manuscript, in both the 2004 and 2015 Canadian Community Health Survey (CCHS) cohorts, participants aged 19 years and older were asked about their current status regarding specific chronic health conditions. It is important to clarify that all these conditions, including osteoporosis and heart disease, were reported by the participants as having been diagnosed by healthcare professionals. Page 3, lines 141-145.

5) Figures 1 and 2: In the legends to Figures 1 and 2, the threshold values of calcium intake and vitamin intake that identify a condition of "inadequacy" must be reported.

Response: Thank you for your comment regarding the threshold values used to define calcium and vitamin D inadequacy in our figures. We have revised the legends of Figures 1 and 2 to include the specific dietary reference intakes as defined by the Institute of Medicine (IOM) Food and Nutrition Board, which served as the basis for identifying inadequacy in our study.

6). Approval of the epidemiological study by the Ethics Committee should be reported.

Response: Thank you for raising the matter of ethics committee approval. We would like to clarify that our study is based on the analysis of secondary data derived from the Canadian Community Health Survey (CCHS), which is publicly available and de-identified to ensure the anonymity of respondents. As such, our research falls under the category of studies that are exempt from ethics approval, as it does not involve direct interaction with human participants or the collection of identifiable personal data. This exemption is in line with standard ethical guidelines for secondary data analysis. We have made a note of this exemption and the nature of the data used in the methods section of our manuscript to ensure transparency and adherence to ethical standards. Page 3, lines 108-111.

Reviewer 3 Report

Comments and Suggestions for Authors

The authors present data on the nutritional an supplemental intake of calcium and vitamin D in Canada in the years 2004 and 2015. The study uses a population based data bank and report the results for patients with osteoporosis,  heart disease, of both conditions. They found a decrease in calcium intake with many patients not reaching the recommended amount, but an increase in vitamin D intake. They discuss, whether the changed nutritional behavior may be the result of increased awareness for  the importance of vitamin D and fear of negative effects of calcium supplements on cardiovascular health.  The data appear solid and are of importance for public health. 

Specific comments: 

It would be interesting to know the data on calcium and vitamin D intake also for the general population (not suffering from osteoporosis or heart disease). Are similar changes also present in the general population?

Do the data in Table 2 represent only patients with osteoporosis and /or heart disease or the general population?

This point should be explained in materials and methods. 

Discussion: ist should be mentioned , that low nutritional intake dose not necessarily represent vitamin D deficiency, because skin production could compensate for low intake

Author Response

Reviewer 3:

The authors present data on the nutritional an supplemental intake of calcium and vitamin D in Canada in the years 2004 and 2015. The study uses a population based data bank and report the results for patients with osteoporosis, heart disease, of both conditions. They found a decrease in calcium intake with many patients not reaching the recommended amount, but an increase in vitamin D intake. They discuss, whether the changed nutritional behavior may be the result of increased awareness for the importance of vitamin D and fear of negative effects of calcium supplements on cardiovascular health.  The data appear solid and are of importance for public health.

Specific comments:

  1. It would be interesting to know the data on calcium and vitamin D intake also for the general population (not suffering from osteoporosis or heart disease). Are similar changes also present in the general population?

Response: Thank you for your valuable comment regarding the inclusion of data on calcium and vitamin D intake in the general population. We acknowledge the importance of understanding these trends in a broader context, beyond just those suffering from osteoporosis or heart disease. To address this, we have indeed conducted separate studies focusing on calcium and vitamin D intake among the general Canadian population. These studies have been published and provide comprehensive insights into the dietary patterns regarding these nutrients over the same time period. We have incorporated references to these studies in the discussion section of our current manuscript. This inclusion is aimed at enriching the discussion by comparing and contrasting the trends observed in the general population with those in the specific cohorts of individuals with osteoporosis or heart disease.

  • Page 14, lines 338-340:
    • Vatanparast, H., Islam, N., Patil, R. P., Shafiee, M., & Whiting, S. J. (2020). Calcium intake from food and supplemental sources decreased in the Canadian population from 2004 to 2015. The Journal of nutrition, 150(4), 833-841.
  • Page 16, lines 421-425:
    • Vatanparast, H., Patil, R. P., Islam, N., Shafiee, M., & Whiting, S. J. (2020). Vitamin D intake from supplemental sources but not from food sources has increased in the Canadian population over time. The Journal of nutrition, 150(3), 526-535.
  1. Do the data in Table 2 represent only patients with osteoporosis and /or heart disease or the general population? This point should be explained in materials and methods.

Response: In the Results section, we specified that Table 2 encompasses data the Canadian population aged 50 years and older, including individuals both with and without diagnoses of osteoporosis and/or heart disease. This table is intended to show the associations between various sociodemographic factors and the likelihood of having osteoporosis or heart disease within this age group. The analysis therefore compares the prevalence and associated risk factors for these conditions across different segments of the population within the specified age range for the years 2004 and 2015. Page 5, lines 226-229.

  1. Discussion: it should be mentioned that low nutritional intake dose not necessarily represent vitamin D deficiency, because skin production could compensate for low intake

Response: We acknowledge the importance of this biological process and its role in overall vitamin D status. In light of your comment, we have added a statement to the discussion section of our manuscript to emphasize that low dietary vitamin D intake does not necessarily indicate a deficiency, given the body's ability to synthesize this nutrient endogenously through skin exposure to UVB radiation. Page 16, lines 440-445.

Reviewer 4 Report

Comments and Suggestions for Authors

Analysis of a large number of adult data from Canadian Health Surveys for Ca and vitamin D intakes collected in 2015 vs 2004 shows declines in Ca intake and increases in vitamin D intakes. A strength of this study is the large representative sample of Canadians which has not been previously used to evaluate the questions.

Could there be a further risk to declining Ca intakes with the change in Canadian food guidance not to specify dairy consumption recommendations?

Author Response

Reviewer 4

Analysis of a large number of adult data from Canadian Health Surveys for Ca and vitamin D intakes collected in 2015 vs 2004 shows declines in Ca intake and increases in vitamin D intakes. A strength of this study is the large representative sample of Canadians which has not been previously used to evaluate the questions.

  1. Could there be a further risk to declining Ca intakes with the change in Canadian food guidance not to specify dairy consumption recommendations?

Response: Thank you for your thoughtful feedback on our manuscript. We revised the conclusion section to address the potential impact of the changes in the food guidelines on calcium intake and underscores the need for public health initiatives to promote dietary diversity to meet nutritional needs. Page 17, lines 479-484.

Round 2

Reviewer 2 Report

Comments and Suggestions for Authors     The Authors responded adequately and comprehensively to all the referee's comments.